# Chronic Kidney Disease and Osteoarthritis: Current Understanding and Future Research Directions

**DOI:** 10.3390/ijms26041567

**Published:** 2025-02-13

**Authors:** Rong-Sen Yang, Ding-Cheng Chan, Yao-Pang Chung, Shing-Hwa Liu

**Affiliations:** 1Department of Orthopedics, College of Medicine and Hospital, National Taiwan University, Taipei 100, Taiwan; rsyang@ntuh.gov.tw; 2Department of Geriatrics and Gerontology, College of Medicine and Hospital, National Taiwan University, Taipei 100, Taiwan; dingchengchan@ntu.edu.tw; 3Institute of Toxicology, College of Medicine, National Taiwan University, Taipei 100, Taiwan; d04447003@ntu.edu.tw; 4Department of Medical Research, China Medical University Hospital, China Medical University, Taichung 406, Taiwan; 5Department of Pediatrics, College of Medicine and Hospital, National Taiwan University, Taipei 100, Taiwan

**Keywords:** chronic kidney disease, osteoarthritis, uremic toxins, iron, calcium

## Abstract

Chronic kidney disease (CKD) is a significant public health concern. Osteoarthritis (OA), a common form of arthritis, has been shown to have a dramatically increased prevalence, particularly among individuals aged 40–50 and older, in the presence of CKD. Furthermore, CKD may exacerbate the progression and impact of OA. A survey study revealed that 53.9% of CKD patients undergoing long-term hemodialysis were diagnosed with OA. These findings underscore the potential association between CKD and OA. Uremic toxins, such as indoxyl sulfate, p-cresyl sulfate, transforming growth factor-β, and advanced glycation end-products, are regarded as potential risk factors in various CKD-related conditions, affecting bone and joint metabolism. However, whether these factors serve as a bridging mechanism between CKD and OA comorbidities, as well as their detailed roles in this context, remains unclear. Addressing the progression of OA in CKD patients and identifying effective treatment and prevention strategies is an urgent challenge that warrants immediate attention. This review focuses on describing and discussing the molecular pathological mechanisms underlying CKD-associated OA and the possible therapeutic strategies.

## 1. Introduction

Chronic kidney disease (CKD) has become a global public health issue, with increasing incidence and prevalence. Many patients progress to kidney failure, requiring kidney transplantation as treatment. Toxic substances and drugs may also contribute to CKD, as seen in conditions like chronic glomerulonephritis and chronic interstitial fibrosis. Certain diseases, such as diabetes and hypertension, are also major causes of CKD. Data from the United States Renal Data System 2023 [1] Annual Data Report indicated that the highest incidence of treated end-stage renal disease (ESRD) in 2021 in the world was in Jalisco, Mexico, followed by Taiwan. Taiwan also had the highest prevalence of both treated ESRD and dialysis among the general population globally in 2021. The U.S. had the highest incidence of kidney transplantation in 2021, followed by Jalisco and Aguascalientes in Mexico, and Israel. The highest incidence of treated ESRD, which was attributed to diabetes mellitus, in 2021 was in Brunei Darussalam, followed by Jalisco (Mexico), Singapore, and Taiwan [1]. According to Taiwan’s National Health Insurance (TNHI) Research Database, kidney diseases, including acute kidney injury (AKI) and CKD, ranked as the highest healthcare expenditure in 2023 [2]. Additionally, the TNHIRD also revealed that joint diseases ranked among the top 20 conditions in healthcare expenditures in 2023. Interestingly, CKD could exacerbate the ranking of other diseases, such as joint disorders.

The hallmark of ESRD is renal interstitial fibrosis, characterized by the replacement of normal kidney structures with abnormally accumulated extracellular matrix proteins, tubular atrophy, tubular dilation, and glomerulosclerosis. Most treatment options for ESRD, including hemodialysis, peritoneal dialysis, and kidney transplantation, are expensive. Among these, only kidney transplantation can restore renal function to near-normal levels. Therefore, developing strategies for the prevention and effective treatment of CKD remains an urgent and critical challenge.

Epidemiological studies showed that during CKD progression, the accumulation of uremic toxins is associated with abnormalities in mineral and bone metabolism, leading to renal osteodystrophy, osteoarthritis (OA), calcific uremic arteriolopathy, and peripheral neuropathy as part of the uremic syndrome [3,4,5,6,7]. As kidney function declines, disturbances in mineral metabolism develop alongside changes in bone structure and strength, thereby increasing the risk of bone disorders in patients with CKD. This condition is referred to as CKD-related mineral and bone disorder (CKD-MBD) or renal osteodystrophy. CKD-MBD is a systemic disease manifested by one or more of the following factors: abnormalities in calcium, phosphorus, parathyroid hormone, or vitamin D metabolism; disturbances in bone turnover, mineralization, volume linear growth, or strength; or extraskeletal calcification [7,8,9]. It results in a loss of bone mineral density and an increased risk of peripheral fractures [6,10,11]. On the other hand, subchondral bone, which provides mechanical support to cartilage and absorbs and distributes mechanical forces, could also be affected by CKD-MBD and increase the risk of OA [4,12].

Uremic toxins, such as indoxyl sulfate (IS), p-cresyl sulfate (PCS), transforming growth factor-beta (TGF-β), and advanced glycation end-products (AGEs), are considered potential risk factors in various CKD-related conditions, including bone and joint metabolism [13,14,15]. AGEs are considered a uremic toxin generated through glycation. The accumulation of AGEs is not merely a consequence of elevated blood glucose levels or reduced renal clearance but also acts as a promoter of CKD progression [16]. Research has indicated a correlation between total IS concentration and AGEs levels in the blood of long-term hemodialysis patients [17]. In a 5/6 nephrectomy-induced uremic rat model, excess IS in the residual nephrons was shown to contribute to increased TGF-β1 bioactivity, enhancing the expression of tissue inhibitor of metalloproteinases (TIMP-1) and type 1 collagen in uremic kidneys. This mechanism accelerates CKD progression [18]. Cheng et al. also reported that IS could upregulate the phosphorylation of signal transducers and activators of transcription-3 (STAT-3), subsequently increasing the production of TGF-β1, MCP-1/CCL2, and α-SMA, which molecular changes are implicated in stromal inflammation, renal fibrosis, and further CKD progression [19].

IS and PCS, in particular, are protein-bound uremic toxins that are difficult to remove through dialysis [20]. Patients undergoing dialysis frequently experience arthritis, osteoporosis, and even fractures, which may be attributed to IS and PCS-induced bone metabolism abnormalities, inflammation, and age-related changes [21]. Moreover, accumulating evidence indicates an increased risk of comorbidity between CKD and OA [4,12]. However, the connection and regulatory mechanisms between CKD and OA remain complex and poorly understood. Identifying effective treatment and prevention strategies is an urgent and critical challenge. Therefore, this review aimed to describe and discuss the current understanding and future research directions, and the regulatory mechanisms underlying CKD-associated OA. The possible therapeutic strategies for CKD-associated OA will be discussed. The search engines and tools from Google Scholar, PubMed, and PubMed Clinical Queries were used to search for the publications that were the subject of this review.

## 2. CKD-Associated OA

OA is the most common form of arthritis, characterized by pain, joint inflammation, and stiffness. It is a complex disease affecting the entire joint cavity [22]. During the progression of OA, changes occur in the integrity of cartilage components, making them more susceptible to physical damage. Initially, cartilage erosion occurs on the joint surface. As the disease progresses, calcification of the cartilage zone expands, cartilage fissures deepen, and there is an increase in matrix degradation products and pro-inflammatory substances. These substances stimulate the adjacent synovium, triggering proliferation and inflammatory responses. Meanwhile, in the subchondral bone, bone remodeling increases, accompanied by vascular invasion penetrating the tidemark into the cartilage [23]. Reactivation of endochondral ossification results in osteophyte formation at joint margins, strongly influenced by inflammatory substances. OA can affect individuals across all age groups, but its prevalence rises sharply among men over 50 and women over 40 [23]. This increase may be associated with the coexistence of other chronic diseases, which can exacerbate OA progression and impact.

OA patients often have additional comorbid risk factors, including hypertension, hypercholesterolemia, low levels of high-density lipoprotein (HDL), renal impairment, and diabetes [24,25,26]. Notably, renal dysfunction and kidney failure are manifestations of CKD [27,28]. Globally, the prevalence of CKD and OA comorbidity is increasing, likely driven by aging populations and associations with comorbidities such as hypertension and obesity [12]. A hospital survey study by Muyodi et al. revealed a CKD prevalence of 61.9% among OA patients, with stage 3A CKD accounting for 45.5% and stage 3B for 13.7%, particularly among individuals over 65 years old [29]. Another study reported that among 34 patients with inflammatory arthritis, 16 also had CKD [30]. Additionally, 53.9% of long-term hemodialysis patients with CKD were diagnosed with OA, and the prevalence increased with longer dialysis duration [31,32]. While these studies indicate a bidirectional risk relationship between CKD and OA, the mechanisms underlying their interaction remain unclear, warranting further investigation.

### 2.1. Role of IS and PCS

CKD is characterized by a gradual loss of kidney function, with a glomerular filtration rate (GFR) less than 60 mL/min/1.73 m^2^. In its later stages, CKD leads to the retention of organic compounds, such as the uremic toxins IS and PCS, and inorganic substances such as phosphate [33]. During CKD progression, the accumulation of protein-bound uremic toxins inhibits skeletal and muscle function and progressively increases the release of calcium and phosphate from bone, exacerbating bone turnover (CKD-MBD). This results in a loss of bone mineral density [8,9]. As kidney function continues to decline, IS accumulates in parallel with sustained secretion of parathyroid hormone (PTH) in the serum [9]. This accumulation enhances resistance to the PTH receptor and increases bone resorption, leading to calcium loss from bones and elevated calcium ion levels in the serum [21,34,35]. Yamamoto and Fukagawa also demonstrated that IS and PCS reduced PTH receptor expression in osteoblasts, impairing bone strength [36]. In addition, IS and PCS can activate inflammatory responses via nuclear factor-κB (NF-κB), stimulating the synthesis of inflammatory cytokines and promoting pro-inflammatory mediators, such as interleukin (IL)-1, IL-6, IL-10, and tumor necrosis factor-α (TNF-α) in vitro and in vivo [37,38,39]. Adesso et al. showed that IS induced NF-κB translocation and free radical release through mitochondrial Ca^2+^ overload, further intensifying inflammation [40]. Moreover, a study by Stockler-Pinto et al. revealed that IS and PCS accumulation in hemodialysis patients correlated with NF-κB upregulation and nuclear factor erythroid 2-related factor 2 (Nrf2) downregulation, leading to inflammation and oxidative stress [41]. Chang et al. further suggested that reducing PCS-induced oxidative stress could mitigate inflammasome activation and inflammatory responses caused by PCS, suggesting a therapeutic strategy for uremic vascular calcification-related cardiovascular events in CKD patients [42].

However, the literature on the effects of IS and PCS on articular chondrocytes or joint function is scarce. A study has indicated that IS may induce inflammatory responses and oxidative stress in synovial fibroblasts, meniscal fibrochondrocytes, and articular chondrocytes [43]. Notably, extensive research has confirmed that CKD promotes inflammation through uremic toxins like IS and PCS. Nevertheless, whether IS and PCS exacerbate the inflammatory processes and progression of CKD-associated OA remains unclear. Therefore, further investigation into IS and PCS as potential molecular mechanisms regulating CKD-associated OA is warranted.

### 2.2. Role of AGEs

The accumulation of AGEs has been linked to complications of diabetes, including CKD. In patients with diabetic nephropathy-induced kidney failure, elevated serum levels of AGEs are frequently observed, along with adynamic bone disease (ABD). AGEs are implicated in the pathogenesis of ABD by inhibiting osteoblast activity and suppressing PTH secretion in response to hypocalcemia [44]. AGEs are a group of compounds formed through non-enzymatic reactions between reducing sugars and amino groups. Their accumulation in bone collagen fibers physically impacts bone quality, and biologically, AGEs act as agonists for their receptor (RAGE), inhibiting bone metabolism [45]. Clinical outcomes in diabetes, such as reduced bone density, suppressed bone turnover markers, and impaired bone mass, may result from AGE–RAGE signaling [46]. Increased AGE formation is not exclusive to diabetes patients; in ESRD patients undergoing hemodialysis, elevated AGE levels are found in blood, skin, and amyloid fibrils [47]. In a rat model of renal osteodystrophy (adenine-induced CKD), AGEs were found to accumulate in osteoblasts, inhibiting their differentiation, reducing lysyl oxidase activity, and inducing bone collagen degradation, leading to fragile bones [48].

Regarding joints and chondrocytes, AGEs accumulation in articular cartilage is considered a major risk factor for OA [49,50,51]. AGE accumulation has been found in aged OA human menisci, suggesting that AGEs may play a role in the meniscus degeneration [50]. Yang et al. found that AGEs induced mitochondrial dysfunction via the inhibition of AMPKα/SIRT1/PGC-1α signaling pathway was involved in the AGE-induced chondrocyte dysfunction [51]. Inhibition in the PPARγ/AMPK/SIRT-1 signaling pathway has been shown to induce inflammatory responses in human articular chondrocyte stimulated by AGEs [49]. Research has demonstrated that hyperglycemia promotes AGE accumulation in rat fibroblast-like synoviocytes via the hypoxia-inducible factor (HIF)-1α-glucose transporter 1 (GLUT1) pathway, increasing the endoplasmic reticulum stress levels and the release of inflammatory factors from synovial cells. This, in turn, induces cartilage degradation and accelerates OA progression [52]. Additionally, in a rat knee joint immobilization model, AGE accumulation, which was induced by ribose injection to increase pentosidine (a type of AGEs) levels, was found to accelerate joint contracture formation [53]. Despite these findings, the specific role of AGEs in CKD-associated OA remains unclear and warrants further investigation.

### 2.3. Role of TGF-β

TGF-β is a pro-fibrotic factor recognized as playing a significant role in the pathogenesis of CKD. It is known as both a positive and negative regulator of cell proliferation, differentiation, and death in various cell types [54]. Elevated levels of active TGF-β have been identified as inducers and promoters of ectopic bone formation, suggesting that TGF-β could serve as a therapeutic target for heterotopic ossification [55]. The TGF-β/bone morphogenetic protein (BMP) pathway plays a crucial role in bone formation during mammalian development and exhibits multiple regulatory functions in the human body; dysregulation of TGF-β/BMP signaling is associated with numerous skeletal disorders in humans [56]. Gene knockout or mutations related to TGF-β or BMP signaling in mice have been shown to result in varying degrees of skeletal abnormalities [56,57]. TGF-β signaling is vital in the differentiation of mesenchymal stem cells (MSCs)/osteoprogenitor cells, osteoblast differentiation, and osteocyte mineralization; the TGF-β–Smad signaling pathway promotes the proliferation, chemotaxis, and early differentiation of osteoprogenitor cells, while it inhibits osteoblast maturation, mineralization, and transition to osteocytes [57]. TGF-β also has dual effects on osteoclastogenesis in that it promotes osteoclast generation by directly binding to receptors on osteoclasts, but inhibits osteoclast differentiation by reducing the RANKL/OPG secretion ratio [58]. TGF-β maintains bone homeostasis through its role in coordinating cellular activities during bone remodeling; however, elevated TGF-β levels in CKD can disrupt mineral metabolism and trigger bone remodeling, contributing to skeletal complications [56].

On the other hand, members of the TGF-β superfamily of secreted factors play essential roles in nearly all aspects of cartilage formation and maintenance [59]. During endochondral ossification, TGF-β acts as a potent inhibitor of terminal differentiation of chondrocytes in the epiphyseal growth plate, primarily mediated through Smad signaling [60]. TGF-β1 has been shown to promote the proliferation of primary murine chondrocytes and extracellular matrix synthesis through the circPhf21a-Vegfa axis, offering a potential therapeutic target for OA [15]. TGF-β3 contributes to the balance between chondrogenic differentiation and chondrocyte hypertrophy, a critical regulatory function in cartilage development. Elevated TGF-β3 levels have a dual role: in healthy tissue, they enhance chondrocyte viability, but in OA cartilage, they may accelerate disease progression [61]. Despite these findings, the precise role of TGF-β in the context of CKD-associated OA remains unclear and requires further investigation.

A summary of the effects on bone versus cartilage by uremic toxins is listed in Table 1.

## 3. Cell Senescence in Joints

OA is a common age-dependent degenerative and inflammatory disease, not merely a simple wear-and-tear issue. Its pathogenesis involves multiple factors, including oxidative stress, apoptosis, senescence, mitochondrial dysfunction, and inflammatory mediators [63,64]. Current evidence suggests that cellular senescence may play a critical role in the inflammatory responses during OA progression [65]. Research has shown that OA chondrocytes and synoviocytes exhibit senescence-related markers, such as increased activity of senescence-associated β-galactosidase (SA-β-Gal) and upregulation of p16 expression [66]. Chondrocytes are considered key players in OA, demonstrating disruptions in the normal balance between extracellular matrix synthesis and degradation during the disease process [66]. A study by Gao et al. revealed a progressive increase in SA-β-Gal expression in cartilage lesions from mild, moderate, to severe knee OA compared to normal cartilage, correlating with disease severity [67]. Ma et al. demonstrated that primary murine articular chondrocytes undergo senescence under inflammatory or oxidative stress conditions [68]. Additionally, Xu et al. found that the antioxidant and anti-inflammatory compound Atractylenolide-III suppressed chondrocyte senescence and matrix metalloproteinase (MMP)-13 production by attenuating the phosphorylation in the NF-κB regulatory pathway, specifically IKKα/β, IκBα, and p65 [69]. Senescent cells also secrete pro-inflammatory cytokines, chemokines, and extracellular matrix proteins, collectively forming a toxic microenvironment known as the senescence-associated secretory phenotype (SASP) [70]. Through SASP secretion, senescent cells can transmit these factors to neighboring bystander cells, potentially inducing more senescent cells and further exacerbating tissue dysfunction [70]. The literature indicates that chondrosenescence, in conjunction with the interaction between inflammatory mediators and an immunosenescent microenvironment, contributes to age-related degeneration of joint cartilage, synovium, and other tissues [71].

In 12-week-old (young control) and 12-month-old C57BL/6 mice subjected to surgically induced OA, transcriptomic analysis of knee joint tissues revealed a significant upregulation of genes involved in Ca^2+^ signaling [72]. Ureshino et al. demonstrated that mitochondrial Ca^2+^ overload during the aging process promoted increased oxidative stress [73]. Similarly, Ren et al. showed that Ca^2+^ influx through the Piezo1 channel protein induced the expression of senescence-related markers such as SA-β-Gal and p16/p21, while treatment with the Ca^2+^ chelator BAPTA reversed cellular senescence [74]. The abnormal Ca^2+^ elevation during senescence further activated calpain proteins, which processed IL-1α precursors into IL-1α and regulated the SASP, including pro-inflammatory secretory components [75].

In a study of 180 healthy individuals, serum concentrations of uremic toxins IS and PCS were significantly higher in patients over 65 years old compared to those under 65 years old, positively correlating with aging and inversely with estimated GFR (eGFR) [76]. IS and PCS accumulation is a hallmark of uremic conditions following CKD-induced renal decline. By altering metabolism, nutrient intake, and gut microbiota dysbiosis, they induce changes in DNA methylation resembling those observed during senescence, promoting aging phenotypes in tissues [77]. Elevated levels of IS and PCS metabolites were observed in the bone tissue of aged (28-month-old) C57BL/6J mice compared to young mice, indicating IS and PCS accumulation during aging [78]. Ribeiro et al. reported that IS promoted moderate inflammation in macrophages (significant increases in MCP1, TNF-α, and IL-10 levels) and senescence in renal tubular epithelial cells (increased percentage of SA-β-Gal-positive cells) via NADPH oxidase 4 (Nox4) upregulation [79]. Similarly, Yang et al. found that IS accelerated senescence in proximal renal tubular cells by upregulating SASP factors (IL-1β, IL-6, and IL-8) through TNF-α and NF-κB signaling pathways [80]. Shimizu et al. showed that reducing IS-induced downstream factors TGF-β and NF-κB expression diminished senescence in proximal renal tubular cells [81]. IS and PCS dose-dependently increased SA-β-Gal and p21 expression in mesenchymal stem cells, participating in senescence and impairing osteoblast mineralization [82,83]. Nevertheless, TGF-β1 has been suggested that it plays an important role in the regulation of chondrocyte characteristics, and the inhibition of the TGF-β1 signaling pathway may lead to senescence and dedifferentiation in chondrocytes [84]. These findings suggest that uremic toxins may play regulatory roles in cellular senescence and inflammation in kidney- and bone-related cells, but their detailed roles and mechanisms in chondrocytes and synoviocytes remain to be clarified.

## 4. Cell Ferroptosis in Joints

Over the past decade, mounting evidence has highlighted the lethal accumulation of unstable iron ions and lipid peroxides as hallmarks of ferroptosis, a process implicated in the pathophysiology of degenerative joint and bone diseases [85]. Ferroptosis can be broadly categorized into three regulatory pathways: (1) The first pathway involves iron metabolism, where ferric ions (Fe^3+^) bind to transferrin (TF) and enter cells via the transferrin receptor (TFR) through endocytosis. Once inside the cell, Fe^3+^ is reduced to ferrous ions (Fe^2+^), which then participate in the Fenton reaction to generate reactive oxygen species (ROS); (2) The second pathway centers on lipid metabolism and involves the activation of lysophospholipid acyltransferase 3 (LPCAT3), long-chain fatty acid CoA ligase 4 (ACSL4), and lipoxygenase (LOX), accelerating the production of lipid peroxides; (3) The third pathway pertains to antioxidant regulation and includes components such as SLC7A11 (System-Xc^−^; cystine/glutamate antiporter), glutathione (GSH), and glutathione peroxidase 4 (GPX4), which work to inhibit peroxidation. When the GSH/GPX4 metabolism is disrupted, coupled with excessive iron and the accumulation of lipid peroxides in cells, lipid peroxyl radicals are rapidly formed, ultimately leading to ferroptosis [85]. These mechanisms work synergistically to drive cell death. Notably, while ferroptosis is a form of cell death, cellular senescence is a survival mechanism. Despite their opposing roles, Wen et al. demonstrated that senescent chondrocytes in OA exhibited highly activated pro-ferroptosis metabolism, including elevated intracellular total iron content, increased ferritin levels, and heightened activities of ACSL4 and LOX [86]. Additionally, both senescence and ferroptosis are characterized by increased oxidative stress, indicating that these processes might co-exist in age-related diseases.

Substantial evidence links ferroptosis to OA and age-related joint degeneration [85,87]. Clinically, patients with hemophilic arthropathy, hemochromatosis arthropathy, and OA show excessive intra-articular iron accumulation [88], with men displaying ferritin levels five times higher than healthy controls [89]. Such iron overload is a hallmark of ferroptosis, characterized by lipid peroxidation, weakened antioxidant defenses, and the release of inflammatory factors like IL-1β and MMP-13, which exacerbate OA [85,90]. In an intervertebral disc degeneration model by bilateral facet joint transection in mice, inhibiting nuclear factor erythroid 2-related factor 2 (Nrf2) may participate in cartilaginous endplate degeneration and ferroptosis of chondrocytes [91]. Similarly, in an anterior cruciate ligament transection (ACLT)-induced OA mouse model, hypoxia-inducible factor 2α (HIF-2α) suppressed the GPX4 antioxidant system and promoted lipid peroxidation, leading to chondrocyte ferroptosis and cartilage degradation [92]. Gong et al. also found that the iron chelator deferoxamine attenuated p53-mediated regulation, upregulating the expression of GPX4, SLC7A11, and collagen II and reducing the expression of MMP-13, iNOS, and COX-2, thereby alleviating OA-related chondrocyte ferroptosis and cartilage degradation [93].

Interestingly, in OA patients, elevated serum iron levels are accompanied by abnormal calcium ion levels; intracellular Ca^2+^ elevation is also a characteristic of ferroptosis [94]. Through transient receptor potential melastatin 7 (TRPM7) or Piezo1 channel proteins, increased intracellular calcium inhibits GPX4, leading to elevated lipid peroxyl radicals and worsening OA severity [95,96]. Therefore, cartilage degradation in OA may not only be linked to the increased Ca^2+^ ions, but also associated with the weakened antioxidant systems by ferroptosis induction. It is worth exploring whether ferroptosis contributes to the progression of CKD-associated OA and the potential interactions between iron and Ca^2+^ ions. Further research into these mechanisms may unveil new insights into the interconnected roles of these ions in degenerative diseases.

## 5. Therapeutic Strategies for CKD-OA

Certain therapeutic strategies used to treat CKD or OA may also be applicable to CKD-associated OA. It is known that there is no effective cure for OA [22]. Traditional treatments for OA include exercise, physical therapy, analgesics for pain, or surgical treatment, including joint replacement. Nonetheless, new therapeutic strategies are still being developed for CKD or OA.

AST-120 (Kremezin): AST-120 is an oral spherical activated carbon that may delay the need for dialysis and alleviate uremic symptoms by adsorbing acidic and alkaline organic compounds, particularly protein-bound uremic toxins [97,98]. Ueda et al. reported that oral administration of AST-120 reduced serum AGEs levels in pre-dialysis CKD patients [99]. Konishi et al. found that early use of AST-120 in patients with overt diabetic nephropathy could prevent the progression of renal dysfunction in type 2 diabetes [100]. Additionally, AST-120 has been shown to exhibit favorable effects in CKD by lowering the serum levels of nephrotoxic metabolites, including AGEs, indoxyl sulfate (IS), and p-cresyl sulfate (PCS) [98,101]. Oral administration of AST-120 has also been reported to improve CKD-MBD by reducing circulating levels of indoxyl sulfate [36]. However, the preventive or therapeutic effects of AST-120 on CKD-associated OA remain unclear.

Anti-senescence drugs for OA: Targeting senescent cells in OA presents a promising strategy for promoting cartilage regeneration. Potential treatments include senolytic agents (inducing apoptosis in senescent chondrogenic progenitor cells), Navitoclax (ABT-263; promoting apoptosis in senescent cells), anti-β-2-microglobulin antibodies (targeting apoptosis in senescent chondrocytes), and rapamycin (enhancing autophagy) [65].

Iron homeostasis drugs or ferroptosis inhibitors for OA: Maintaining iron homeostasis is essential for joint health, as excessive iron can cause oxidative stress and damage, which may contribute to aging-related diseases. Chondrocyte ferroptosis has been suggested as a therapeutic target in OA [90]. There are several drugs that help regulate iron homeostasis or inhibit ferroptosis, such as Deferoxamine (a chelating agent for iron), Lactoferrin (a non-haem iron-binding protein), Icariin (a flavonoid glycoside), Resveratrol (a natural phenolic compound), Ferrostatin-1 (an antioxidant; ferroptosis inhibitor), N-acetylcysteine (an antioxidant; ferroptosis inhibitor), and others [88].

Miscellaneous therapeutic strategies in OA: There are various new treatment strategies being developed for OA, including symptom modifiers (such as liposome-based dexamethasone, microspore-based triamcinolone, and nerve growth factor antagonist); disease-modifying agents [such as anti-ADAMTS-5 (a disintegrin and metalloproteinase thrombospomdin motifs-5), pentosan polysulfate sodium, allogeneic stem cells, C-C chemokine receptor type-4 (CCR4) ligand 17 inhibitor, and Wnt-signaling inhibitors]; anti-obesity [Retatrutide (LY3437943), an agonist of the glucose-dependent insulinotropic polypeptide (GIP), glucagon-like peptide-1 (GLP-1), and glucagon receptors, for knee/hip OA with obesity]; genicular nerve block (with betamethasone and bupivacaine or radio-frequency ablation) [102].

On the other hand, non-steroidal anti-inflammatory drugs (NSAIDs), such as celecoxib and diclofenac, are commonly recommended as first-line treatments for OA-related inflammatory pain and physical dysfunction [103,104]. NSAIDs may also be used for CKD-associated OA; however, long-term and consistent use carries an increased risk of nephrotoxicity, particularly in patients with pre-existing CKD [103,105]. Potential nephrotoxic effects of NSAIDs include acute kidney injury (AKI), CKD progression, tubulointerstitial nephritis, papillary necrosis, pre-renal azotemia, and acute tubular necrosis [105,106]. Therefore, NSAID use in CKD-associated OA patients should be approached with caution, balancing pain relief with potential kidney-related risks.

## 6. Conclusions and Perspectives

Valid biomarkers are essential tools for the early diagnosis and therapeutic intervention of CKD and its associated complications. Traditional biomarkers for CKD, such as creatinine, eGFR, and urinary albumin-to-creatinine ratio (uACR), remain widely used. In addition, novel biomarkers, including neutrophil gelatinase-associated lipocalin (NGAL), cystatin C, beta-2 microglobulin (B2M), beta trace protein (BTP), klotho, symmetrical dimethylarginine (SDMA), and dickkopf-3 (DKK3), have shown promise in CKD assessment [107,108,109]. Certain biomarkers, such as parathyroid hormone (PTH), serum calcium and phosphate, fibroblast growth factor 23 (FGF23), 1,25-dihydroxyvitamin D (1,25(OH)2D), and sclerostin, have been suggested for clinical evaluation of CKD-MBD [110,111]. Recently, emerging biomarkers such as adropin, a secreted peptide, and afamin, a vitamin E-binding glycoprotein, have been investigated for their potential role in monitoring CKD progression, including CKD-MBD [112,113,114]. However, there are currently no specific biomarkers for CKD-associated OA. Interestingly, studies have reported a decrease in serum adropin levels in knee OA patients with a body mass index (BMI) above 30, suggesting that adropin may serve as a novel biomarker for knee OA [115]. Nevertheless, further research is needed to develop specific biomarkers for CKD-associated OA.

An increasing body of evidence suggests a heightened risk of comorbidity between CKD and OA. However, the underlying connections and regulatory mechanisms remain complex and poorly understood. This review explored potential risk factors, including IS, PCS, TGF-β, and AGEs, which influence bone and joint metabolism in CKD-related conditions. Uremic toxins may play a crucial pathological role in CKD-associated OA. Additionally, we discussed the potential contributions of cellular senescence and ferroptosis to joint/cartilage degeneration under CKD conditions. Excessive iron and calcium accumulation may trigger oxidative stress and inflammatory responses, potentially accelerating OA progression [85,90]. A recent study suggested that elevated serum AGEs levels in osteoporosis patients could induce ferroptosis in osteoblasts, thereby promoting osteoporosis [62]. However, the role of ferroptosis in uremic toxin-induced joint cell damage and CKD-associated OA remains to be elucidated.

A schematic model illustrating potential signaling pathways involved in CKD-associated OA is shown in Figure 1. There are three possible mechanisms regulating cellular ferroptosis and senescence in OA joints under CKD conditions with uremic toxin accumulation: (1) transferrin (Tf)/Tf receptor-mediated iron metabolism, (2) the glutamate-cystine antiporter (Xc^−^)-regulated antioxidant system, and (3) the Piezo1/transient receptor potential melastatin 7 (TRPM7)-mediated calcium-related signaling pathway. Understanding the interplay between uremic toxins, cellular senescence, and ferroptosis may provide new insights into the molecular mechanisms underlying CKD-associated OA and inform potential therapeutic strategies.

## Figures and Tables

**Figure 1 ijms-26-01567-f001:**
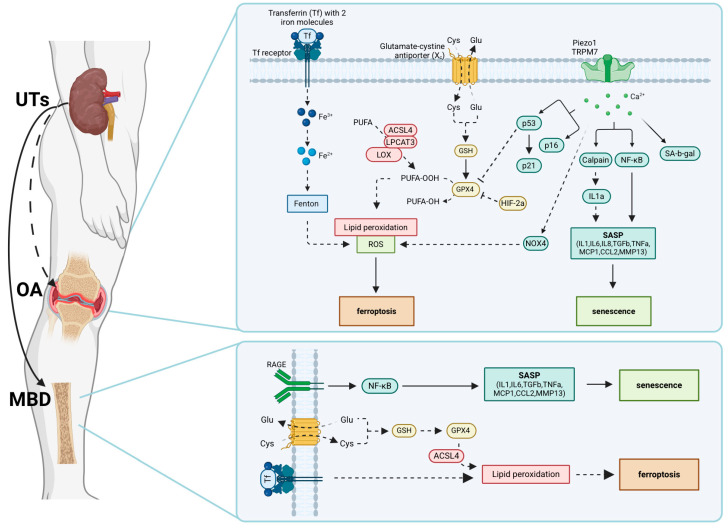
The accumulation of uremic toxins (UTs) during chronic kidney disease (CKD) disrupts the intracellular balance of iron and calcium ions in joint cells, such as chondrocytes and synoviocytes, leading to ferroptosis and senescence, ultimately causing osteoarthritis (OA). In addition, the CKD-MBD-related signaling pathways are used for comparison. UTs promote iron uptake, increasing intracellular iron ions, which in turn trigger the Fenton reaction to enhance ROS. This inhibits the antioxidant system and exacerbates lipid peroxidation. Additionally, the increase in intracellular calcium ions induces senescence and activates the Calpain and NF-κB signaling pathways, releasing the senescence-associated secretory phenotype (SASP). Solid lines indicate established pathways. Dashed lines represent signaling pathways that have been speculated but not fully substantiated. Long-chain fatty acid CoA ligase 4, ACSL4; lysophospholipid acyltransferase 3, LPCAT3; lipoxygenase, LOX; polyunsaturated fatty acids, PUFA; cystine, Cys; glutamate, Glu; glutathione, GSH; glutathione peroxidase 4, GPX4; hypoxia-inducible factor 2α, HIF-2α; Piezo-type mechanosensitive ion channel component 1, Piezo1; transient receptor potential melastatin 7, TRPM7; nuclear factor-κB, NF-κB; senescence-associated β-galactosidase, SA-β-Gal; NADPH oxidase 4, Nox4; reactive oxygen species, ROS; interleukin, IL; transforming growth factor-beta, TGF-β; tumor necrosis factor-α, TNF-α; metalloproteinase, MMP; monocyte chemoattractant protein-1, MCP1; chemokine ligand 2, CCL2. Created with BioRender.com.

**Table 1 ijms-26-01567-t001:** Summary for affecting bone versus cartilage by uremic toxins.

Uremic Toxins	Bone	Cartilage
IS/PCS	IS/PCS accumulation and lost of bone mineral density during CKD progression in epidemiological studies [8,9]; enhanced resistance to the PTH receptor and increased bone resorption, leading to calcium loss from bones and elevated serum calcium [21,34,35]; reduced PTH receptor expression in osteoblasts, impairing bone strength [36].	IS induced inflammatory responses and oxidative stress in synovial fibroblasts, meniscal fibrochondrocytes, and articular chondrocytes in vitro [43]
AGEs	AGEs inhibited osteoblast activity in a human osteoblastic cell line [44]; the AGE/RAGE axis downregulated bone metabolism, impairing bone mass [45,46]; AGEs accumulated in osteoblasts, inhibiting their differentiation and inducing bone collagen degradation, leading to fragile bones in a rat model of renal osteodystrophy [48]; inhibited proliferation, differentiation, and mineralization in osteoblasts via ferroptosis induction in a human osteoblast cell line [62].	AGEs induced inflammation in human articular chondrocytes via a PPARγ-related pathway [49]; induced mitochondrial dysfunction, ROS production, and NF-κB activation, contributing to dysfunction in human articular chondrocytes [51]; induced inflammatory responses in human articular chondrocytes [49]; hyperglycemia promoted AGE accumulation in the synovium of OA patients and OA rats and in rat fibroblast-like synoviocytes, accelerating the release of inflammatory factors, endoplasmic reticulum stress, cartilage degradation, and OA progression [52]; accelerated joint contracture formation in a rat knee joint immobilization model [53]
TGF-β	Elevated TGF-β could be the inducer and promoter of ectopic bone formation in a mouse model [55]; TGF-β/BMP dysregulation is associated with numerous skeletal disorders in humans and animals [56,57]; TGF-β promoted osteoclast generation but inhibited osteoclast differentiation [58].	TGF-β inhibited terminal differentiation of chondrocytes in the epiphyseal growth plate during endochondral ossification [60]; promoted chondrocyte proliferation and extracellular matrix synthesis in primary mouse chondrocytes and a mouse OA model [15]; elevated TGF-β3 levels enhanced chondrocyte viability in healthy tissue, but accelerated disease progression in OA cartilage [61]

AGEs: advanced glycation end-products; CKD: chronic kidney disease; IS: indoxyl sulfate; OA: osteoarthritis; PCS: p-cresyl sulfate; PPARγ: peroxisome proliferator-activated receptor-γ; PTH: parathyroid hormone; RAGE: receptor for AGE; ROS: reactive oxygen species; TGF-β: transforming growth factor-β.

## Data Availability

No new data were created.

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
