# Peer review of "Chronic Kidney Disease and Osteoarthritis: Current Understanding and Future Research Directions"

_ijms, 2025, doi:10.3390/ijms26041567_

Round 1
Reviewer 1 Report
Comments and Suggestions for Authors
This review looks at the link between Chronic Kidney Disease (CKD) and Osteoarthritis (OA), focusing on how these two conditions might be connected at the cellular and molecular level. It also discusses possible treatments. The article does a good job explaining how things like uremic toxins (waste products that build up in the body during kidney disease), cell aging, and a type of cell death called ferroptosis, might damage joints. These explanations are supported by helpful diagrams. However, there are a few things that could make this review even stronger:
1. The abstract mentions specific percentages (53.9% and 61.9%) regarding how often OA occurs in people with CKD. It's unclear if these numbers come from the same study, which is confusing.
2. The review talks about both bone and cartilage problems in CKD, but doesn't always clearly separate them. Osteoarthritis mainly affects cartilage, but some of the evidence discussed is actually about bone. It's important to be clear about which tissue is being affected by the different mechanisms discussed. The diagram (Figure 1) showing the cellular pathways should also be clearer about whether it's showing what happens in bone or cartilage. The diagram should also be updated to show which pathways are proven and which are just guesses for now. Alternatively, the figure could have two parts: one for confirmed mechanisms and one for proposed ones.
3. A detailed table would be very helpful. This table should clearly show how uremic toxins and different molecular pathways affect bone versus cartilage. It should include: (1) the specific cell types involved (like bone-forming cells or cartilage cells), (2) the molecules involved, (3) the pathways these molecules activate, and (4) references to the scientific studies that found this information.
4. The authors need to add a short paragraph summarizing the types of studies they cited. For instance, they could mention that only four in vitro studies directly tested uremic toxins on cartilage cells (e.g., Chen et al. [43]), and only one animal study was used (Julovi et al. [12]). This will help readers quickly understand what kind of evidence is available and what is still missing. This concise summary would increase the evidence base.
5. The current title, "Chronic Kidney Disease and Osteoarthritis: Up-to-Date Evidence," makes it sound like there's a lot of strong evidence connecting these diseases. However, there isn't much direct proof that CKD-related toxins directly harm cartilage. A better title might be "Chronic Kidney Disease and Osteoarthritis: Current Understanding and Future Research Directions" or "Chronic Kidney Disease and Osteoarthritis: Emerging Concepts and Knowledge Gaps."
Author Response
Reviewer 1
This review looks at the link between Chronic Kidney Disease (CKD) and Osteoarthritis (OA), focusing on how these two conditions might be connected at the cellular and molecular level. It also discusses possible treatments. The article does a good job explaining how things like uremic toxins (waste products that build up in the body during kidney disease), cell aging, and a type of cell death called ferroptosis, might damage joints. These explanations are supported by helpful diagrams. However, there are a few things that could make this review even stronger:
- The abstract mentions specific percentages (53.9% and 61.9%) regarding how often OA occurs in people with CKD. It's unclear if these numbers come from the same study, which is confusing.
Response: Thanks the reviewer’s comment. We're sorry for the confusion. We have revised the descriptions in this revised manuscript according to the suggestion of reviewer. To avoid confusion, we deleted the description of “while the overall prevalence of CKD among OA patients reached 61.9%”.
- The review talks about both bone and cartilage problems in CKD, but doesn't always clearly separate them. Osteoarthritis mainly affects cartilage, but some of the evidence discussed is actually about bone. It's important to be clear about which tissue is being affected by the different mechanisms discussed. The diagram (Figure 1) showing the cellular pathways should also be clearer about whether it's showing what happens in bone or cartilage. The diagram should also be updated to show which pathways are proven and which are just guesses for now. Alternatively, the figure could have two parts: one for confirmed mechanisms and one for proposed ones.
Response: Thanks the reviewer’s comment. We have revised the manuscript for this issue according to the suggestion of reviewer. In this revised Figure 1, we showed two parts, one for cartilage and another for bone. Moreover, solid lines indicate established pathways and dashed lines represent signaling pathways that have been speculated but not fully substantiated.
- A detailed table would be very helpful. This table should clearly show how uremic toxins and different molecular pathways affect bone versus cartilage. It should include: (1) the specific cell types involved (like bone-forming cells or cartilage cells), (2) the molecules involved, (3) the pathways these molecules activate, and (4) references to the scientific studies that found this information.
Response: Thanks the reviewer’s comment. We have prepared a table for summary of affecting bone versus cartilage by uremic toxins (Table 1) in this revised manuscript according to the suggestion of reviewer.
- The authors need to add a short paragraph summarizing the types of studies they cited. For instance, they could mention that only four in vitro studies directly tested uremic toxins on cartilage cells (e.g., Chen et al. [43]), and only one animal study was used (Julovi et al. [12]). This will help readers quickly understand what kind of evidence is available and what is still missing. This concise summary would increase the evidence base.
Response: Thanks the reviewer’s comment. We have added the information for this issue in this revised manuscript according to the suggestion of reviewer. We also summarize these information in Table 1.
- The current title, "Chronic Kidney Disease and Osteoarthritis: Up-to-Date Evidence," makes it sound like there's a lot of strong evidence connecting these diseases. However, there isn't much direct proof that CKD-related toxins directly harm cartilage. A better title might be "Chronic Kidney Disease and Osteoarthritis: Current Understanding and Future Research Directions" or "Chronic Kidney Disease and Osteoarthritis: Emerging Concepts and Knowledge Gaps."
Response: Thanks the reviewer’s comment. We agree with the reviewer's opinion. We have changed the title to the “Chronic Kidney Disease and Osteoarthritis: Current Understanding and Future Research Directions” according to the suggestion of reviewer.
Reviewer 2 Report
Comments and Suggestions for Authors
Rong-Sen Yang submitted a manuscript titled Chronic Kidney Disease and Osteoarthritis: Up-to-Date Evidence. The manuscript is a review. In this review, the authors focused on the molecular pathological mechanisms underlying CKD-associated OA and the possible therapeutic strategies.
The topics described covered several factors that correlate CKD and OA, and include the Role of Indoxyl Sulfate (IS) and p-Cresyl Sulfate (PCS), Advanced Glycation End Products (AGEs) and Transforming Growth Factor (TGF)-β. They then addressed aspects related to OA, in particular Cell Senescence and Cell Ferroptosis. Finally the authors concluded their review by describing the possible therapeutic strategies for the treatment of CKD-OA.
I have only a few suggestions for the authors
Please:
-describe the methodology used to search for the publications that were the subject of this review.
-indicate the terms with their respective acronym only the first time (Advanced glycation end products; indoxyl sulfate).
-Introduce a summary figure of the aspects related to the described markers
-In addition, the authors can make a short paragraph on the future aspects of research oriented to the search for new and more recent biological markers, although little studied, correlating the CKD with OA, subject of future studies, such as Adropin, Afamin, and others.
Author Response
Reviewer 2
Rong-Sen Yang submitted a manuscript titled Chronic Kidney Disease and Osteoarthritis: Up-to-Date Evidence. The manuscript is a review. In this review, the authors focused on the molecular pathological mechanisms underlying CKD-associated OA and the possible therapeutic strategies.
The topics described covered several factors that correlate CKD and OA, and include the Role of Indoxyl Sulfate (IS) and p-Cresyl Sulfate (PCS), Advanced Glycation End Products (AGEs) and Transforming Growth Factor (TGF)-β. They then addressed aspects related to OA, in particular Cell Senescence and Cell Ferroptosis. Finally the authors concluded their review by describing the possible therapeutic strategies for the treatment of CKD-OA.
I have only a few suggestions for the authors
Please:
(1) -describe the methodology used to search for the publications that were the subject of this review.
Response: Thanks the reviewer’s comment. We have added the information of search engines and tools for the publications of this review in this revised manuscript (the end of the Introduction section) according to the suggestion of reviewer.
(2) -indicate the terms with their respective acronym only the first time (Advanced glycation end products; indoxyl sulfate).
Response: Thanks the reviewer’s comment. We have indicated the terms with their respective acronym only the first time in this revised manuscript according to the suggestion of reviewer.
(3) -Introduce a summary figure of the aspects related to the described markers
Response: Thanks the reviewer’s comment. We have added the information for this issue in the Table 1 and the Conclusions and Perspectives section of this revised manuscript according to the suggestion of reviewer.
(4) -In addition, the authors can make a short paragraph on the future aspects of research oriented to the search for new and more recent biological markers, although little studied, correlating the CKD with OA, subject of future studies, such as Adropin, Afamin, and others.
Response: Thanks the reviewer’s comment. We have added the information for new and more recent biomarkers for CKD and its associated complications in the Conclusions and Perspectives section of this revised manuscript according to the suggestion of reviewer.
Reviewer 3 Report
Comments and Suggestions for Authors
The manuscript entitled ‘Chronic Kidney Disease and Osteoarthritis: Up-to-Date Evidence' addresses the interesting issue of chronic kidney disease in patients with osteoarthritis. Corrections are required before the article is eligible for publication:
1. A graphic abstract is recommended.
2. The authors should explain what new contributions their article makes to the area of knowledge under discussion.
3. Markers of renal injury and advances in knowledge in this area should be characterized.
4. The authors should discuss the use of non-steroidal anti-inflammatory drugs in patients with osteoarthritis, in the context of the risk of renal damage.
Author Response
Reviewer 3
The manuscript entitled ‘Chronic Kidney Disease and Osteoarthritis: Up-to-Date Evidence' addresses the interesting issue of chronic kidney disease in patients with osteoarthritis. Corrections are required before the article is eligible for publication:
- A graphic abstract is recommended.
Response: Thanks the reviewer’s comment. We have added a graphic abstract in this revised manuscript according to the suggestion of reviewer.
- The authors should explain what new contributions their article makes to the area of knowledge under discussion.
Response: Thanks the reviewer’s comment. We have added the information for this issue in the Introduction and the Conclusions and Perspectives sections of this revised manuscript according to the suggestion of reviewer.
- Markers of renal injury and advances in knowledge in this area should be characterized.
Response: Thanks the reviewer’s comment. We have added the information for new and more recent biomarkers for CKD and its associated complications in the Conclusions and Perspectives section of this revised manuscript according to the suggestion of reviewer.
- The authors should discuss the use of non-steroidal anti-inflammatory drugs in patients with osteoarthritis, in the context of the risk of renal damage.
Response: Thanks the reviewer’s comment. We have added the information for this issue in the end of Therapeutic strategies for CKD-OA section of this revised manuscript according to the suggestion of reviewer.
Round 2
Reviewer 1 Report
Comments and Suggestions for Authors
The authors have addressed all comments and suggestions. The abstract has been clarified, and Figure 1 now clearly distinguishes between bone and cartilage with established and speculative pathways. A detailed table has been added, summarizing the types of studies and referencing the sources. The title has also been revised to better reflect the manuscript’s content. These revisions have significantly improved the manuscript, which is now suitable for publication.
Reviewer 2 Report
Comments and Suggestions for Authors
No further comments
Reviewer 3 Report
Comments and Suggestions for Authors
The authors have corrected the manuscript and I recommend it for further proceedings.